

# Heavy metal movement through insect food chains in pristine thermal springs of Yellowstone National Park

Braymond Adams[1], John Bowley[2], Monica Rohwer[3], Erik Oberg[4], Kelly Willemssens[3], Wendy Wintersteen[1], Robert K.D. Peterson[2] and Leon G. Higley[3]

[1] Department of Plant Pathology, Entomology, and Microbiology, Iowa State University, Ames, IA, United States of America
[2] Department of Land Resources & Environmental Sciences, Montana State University, Bozeman, MT, United States of America
[3] School of Natural Resources, University of Nebraska-Lincoln, Lincoln, NE, United States of America
[4] Yellowstone National Park, Gardner, MT, United States of America

Corresponding author
Leon G. Higley,
lhigley@drshigley.com

## ABSTRACT

Yellowstone National Park thermal features regularly discharge various heavy metals and metalloids. These metals are taken up by microorganisms that often form mats in thermal springs. These microbial mats also serve as food sources for invertebrate assemblages. To examine how heavy metals move through insect food webs associated with hot springs, two sites were selected for this study. Dragon-Beowulf Hot Springs, acid-sulfate chloride springs, have a pH of 2.9, water temperatures above 70 °C, and populations of thermophilic bacterial, archaeal, and algal mats. Rabbit Creek Hot Springs, alkaline springs, have a pH of up to 9, some water temperatures in excess of 60 °C, and are populated with thermophilic and phototrophic bacterial mats. Mats in both hydrothermal systems form the trophic base and support active metal transfer to terrestrial food chains. In both types of springs, invertebrates bioaccumulated heavy metals including chromium, manganese, cobalt, nickel, copper, cadmium, mercury, tin and lead, and the metalloids arsenic, selenium, and antimony resulting from consuming the algal and bacterial mat biomass. At least two orders of magnitude increase in concentrations were observed in the ephydrid shore fly *Paracoenia turbida*, as compared to the mats for all metals except antimony, mercury, and lead. The highest bioaccumulation factor (BAF) of 729 was observed for chromium. At the other end of the food web, the invertebrate apex predator, *Cicindelidia haemorrhagica*, had at least a 10-fold BAF for all metals at some location-year combinations, except with antimony. Of other taxa, high BAFs were observed with zinc for *Nebria* sp. (2180) and for *Salda littoralis* (1080). This accumulation, occurring between primary producer and primary consumer trophic levels at both springs, is biomagnified through the trophic web. These observations suggest trace metals enter the geothermal food web through the microbial mat community and are then transferred through the food chain. Also, while bioaccumulation of arsenic is uncommon, we observed five instances of increases near or exceeding 10-fold: *Odontomyia* sp. larvae (13.6), *P. turbida* (34.8), *C. haemorrhagica* (9.7), *Rhagovelia distincta* (16.3), and *Ambrysus mormon* (42.8).

# INTRODUCTION

Heavy metals (naturally occurring elements with a high atomic weight at least 5 times greater than that of water) are defined as any member of a subset of elements that exhibit metallic properties—including the transition metals, some metalloids, lanthanides, and actinides (*Tchouwou et al., 2012*). These metals may include the common transition metals, such as copper, lead, and zinc, which are often a cause of environmental pollution from sources such as leaded petroleum, industrial effluents, and drainage from abandoned mines (*Timothy & Williams, 2019*).

Yellowstone National Park (YNP) geothermal geysers, fumaroles, vents, and hot springs emit into the atmosphere gaseous and aqueous solute forms of mercury, sulfur, and other heavy metals (*Bennett & Wetmore, 1999*). Heavy metals are deposited in the adjacent soil through soil degassing and emitted solutions, with emissions varying among thermal vents. For example, arsenic, fluoride, sodium, sulfate, lead, copper, selenium, and mercury all occur in Old Faithful (*National Park Service, 2020*). Although some of these metals are potentially toxic to animals (*e.g.*, arsenic, copper, and methylmercury; *Inskeep & McDermott, 2005*), there is little known about how animals living within and around these thermal features tolerate exposures.

As insects are the most common animals in thermal areas, they are of particular interest for understanding metal impacts and movements from thermal vents. Although a body of literature is developing that shows insects such as chironomids and hydropsychid caddisflies are relatively metal tolerant, other insects, for example many mayflies, particularly within the family Heptageniidae, are highly sensitive to metals (*Cain, Luoma & Wallace, 2004*). Some metal tolerance mechanisms are generally understood, but their expression in stream insects and significance to population and community-level effects have not been established (*Cain, Luoma & Wallace, 2004*).

To measure metal uptake, bioaccumulation is determined. A bioaccumulation factor (BAF) is the concentration of a substance in the biomass of an organism divided by the concentration of that substance within the biomass of a food web's primary producer. If BAFs for a compound increase with trophic level, the compound is said to biomagnify (*Bertato, Chirico & Papa, 2023*). An assessment of biomagnification in a food web can help determine levels of environmental exposure to species of concern as well as elucidate trophic relationships.

At YNP, research on bioaccumulation has focused on mercury. *Boyd et al. (2009)* documented bioaccumulation of total and methylated mercury in tracheal tissue of stratiomyid soldier fly larvae sampled from the thermal outflow channel at Dragon-Beowulf Hot Springs in the Norris Geyser Basin. Methylmercury was shown to bioaccumulate in soldier fly larvae relative to their algal mat food sources, and in avian species (*i.e.,* killdeer) as a result of predation on those larvae (*Braune, 1987*; *Boyd et al., 2009*; *Hurtado et al., 2023*). As an example, killdeer (Charadriiformes: Charadriidae) *Charadrius vociferous*, are known to feed on ephydrid (Diptera: Ephydridae) (shore fly) and stratiomyid (Diptera: Stratiomyidae) larvae that inhabit thermal springs of YNP (*Burger, 1993*; *Boyd et al., 2009*). A molted feather discarded by a killdeer was recovered and analyzed for

methylmercury (*Burger, 1993*; *Boyd et al., 2009*). The feather contained 202 ng/g-dry weight methylmercury, which was a sixfold enrichment relative to the levels in the larval tissue and approaching the concentration (400 ng/g dry weight) accepted as threshold for protection of fish-eating birds (*Mora et al., 2002*).

In addition to ephydrids and stratiomyids, the tiger beetle, *Cicindelidia haemorrhagica*, (Coleoptera: Carabidae), lives in and around thermal pools in YNP, and exhibits striking levels of high thermal tolerance (>50 °C) (*Willemssens, 2019*). This species also demonstrates tolerance to heavy metals. *Gotschall (2021)* found that *C. haemorrhagica* mobilizes various heavy metals into its exoskeleton and cuticular waxes, which is potentially an important mechanism for detoxification. What is not clear from this previous work is if *C. haemorrhagica* and other insects in the thermal pool communities are bioaccumulating and biomagnifying heavy metals, and if concentrations of heavy metals occur at potentially toxic levels. Because insects here are at the base of some vertebrate food chains (birds, small mammals, and reptiles) (*e.g.*, *Boyd et al., 2009*), if *C. haemorrhagica* and other insects are bioaccumulating heavy metals understanding contamination of these insects provides an essential foundation for broader characterizations of heavy metal movement.

Consequently, the objectives of our study were to sample the insects at two hot springs in YNP, measure the occurrence of heavy metals in various insect species, determine if biomagnification is occurring and if so, at what level.

## METHODS AND MATERIALS

### Site descriptions

Studies were carried out under YNP Research Permits #8100 and #7092. Rabbit Creek, YNP (WY, USA), is an alkaline stream near the lower loop of the Midway Geyser Basin in the western area of YNP; Dragon-Beowulf Hot Springs, YNP (WY, USA), are acid-sulfate-chloride springs in the One Hundred Springs area of Norris Geyser Basin (Fig. 1). The sampling sites for the two springs were selected based on ease of access, observable beetle food sources, overall beetle abundance, spring water temperature, and spring water pH.

Dragon-Beowulf Hot Springs are classified as acid-sulfate-chloride-type springs (*Langner et al., 2001*) due to a distinctive chemical signature that includes significant acidity and high concentrations of sulfate and chloride. These springs are relatively well-studied and feature *Hydrogenobaculum*-like Bacteria (*D'Imperio et al., 2007*; *Boyd et al., 2009*) and sulfur metabolizing Archaea in the higher temperature zones and thermophilic photoautotrophic algae once the temperatures drop enough to permit photosynthetic metabolism. At even lower temperatures (<32C), Dragon-Beowulf Hot Springs support the growth of acidophilic phototroph *Cyanidioschyzon* spp. algal mats, which in turn serve as food for stratiomyid and ephydrid populations (*Boyd et al., 2009*; *Zack, 1983*). Dragon-Beowulf Hot Springs source waters originate from the subsurface and have a pH of 2.8 that varies seasonally with rainfall, indicating a relatively shallow source. Insect taxa were collected in this habitat over a 100-m path along the margin of the stream most reflective of *C. haemorrhagica* activity.

Rabbit Creek is an alkaline stream near the lower loop of the Midway Geyser Basin. The main water flow originates from one alkaline hot spring on the eastern edge of

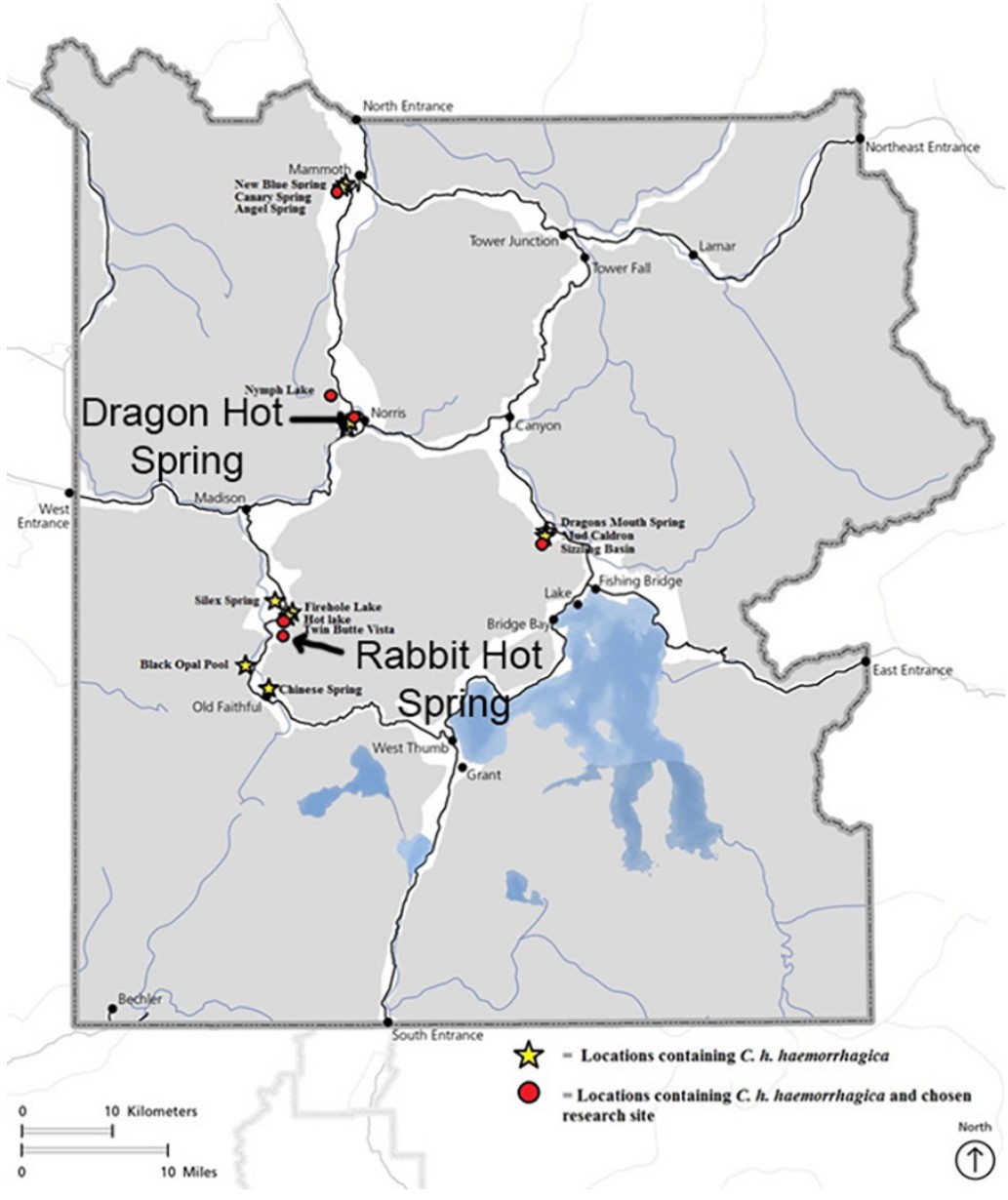

**Figure 1** Map of Yellowstone National Park indicating *Cicindelidia haemorrhagica* habitat and research locations including Dragon-Beowulf Hot Springs and Rabbit Creek Hot Spring. The research sites are indicated on the map with arrows.

the hydrothermal area (buttresses the Mallard Lake Dome) and contains deposits of alkaline siliceous rock throughout most of the water's edge and creek bed (*Bowley, 2021*). Rabbit Creek Hot Springs features alkaline thermal sources with source water temperatures in excess of 60 °C (*Bowley, 2021*). Although the upper portions are adjacent to meadow, some pools and reaches of the spring are completely surrounded by forest. These pools accommodate acidobacteria and cyanobacterial mats (*Weltzer & Miller, 2013*;

*Miller et al., 2009*). A thermal gradient flows throughout the entire reach with estimated changes in water temperature of 7.5 °C for every 100 m of continuous waterway (*Bowley, 2021*). This area has the second largest concentration of hot springs in the basin and flows into the eastern tributary of the Firehole River (*Miller et al., 2009*). With a pH of 9 in Rabbit Creek Hot Springs in 2022 (B Adams, pers. obs., 2021), the presence of *C. haemorrhagica* indicated pH tolerance for the water in which they forage. Additional insect taxa were collected in these habitats; their presence also suggests tolerance to the extreme pH and temperatures.

## Sampling

In July 2021 and 2022, mat biomass and insect taxa were collected from several functional feeding groups, including collector-gatherers, grazers, shredders, and predators, along a 100-m path at four sites over two days in each year (Figs. 2 and 3). *Cyanidioscyzon* spp. algal and cyanobacterial mat biomass represented the base of the food webs at Dragon-Beowulf and Rabbit Hot Springs, respectively. *Paracoenia turbida* (Diptera: Ephydridae), representing the second trophic level, occurred at both locations. *Odontomyia* sp. (Diptera: Stratiomyidae) were collected exclusively at Dragon-Beowulf Hot Springs and also represented the second trophic level. The only predaceous (third trophic level) insect species found at Dragon-Beowulf Hot Springs was *Nebria* sp. (Coleoptera: Carabidae). Predaceous (third trophic level) insects found at Rabbit Creek Hot Springs were *Ambrysus mormon* (Heteroptera: Naucoridae) and *Rhagovelia distincta* (Heteroptera: Veliidae). The presumed apex invertebrate predator at both sites was the tiger beetle *C. haemorrhagica* (Coleoptera: Carabidae) (third or fourth trophic level).

Insects were identified to family and species. Identifications were made with keys appropriate to specific taxa. For Carabidae: *Kavanaugh (1978)*; for Cicindelidae: *Pearson, Knisley & Kozilek (2005)* and *Gough et al. (2018)*; for Ephyrididae: *Brues (1924)* and *Mathis (1975)*; for Naucoridae: *Usinger (1941)*; for Saldidae: *Drake & Hottes (1950)*; for Stratiomyidae: *James (1981)*; and for Vellidae: *Smith & Polhemus (1978)*.

*Cicindelidia haemorrhagica* and *Odontomyia* sp. (Diptera) adults were collected using a sweep net. Larval (sixth instar) *Odontomyia* sp. were collected with forceps. *Paracoenia turbida* adults were collected using sweep net, flight interception, passive flight traps, and aspiration. Adult *Salda littoralis* and *Nebria* sp. were collected with aspiration. Adult *Ambrysus mormon* and *Rhagovelia distincta* were collected using a catch pan and forceps. These collection methods did not preclude the presence of dust on the individual insects, but given their size and relative lack of adhesive surfaces (*e.g.,* abundant setae) we think the likelihood of sample contamination from dust is negligible.

After collection, insects and mat samples were transferred to collection vials, which were then kept on ice (ca. 0 °C) in a portable Yeti Hopper M30 Soft Cooler until they were brought back to Montana State University. In the laboratory, vials were fitted with an airtight lid and stored at −20 °C until processed.

All organisms were placed in acid-washed polypropylene scintillation vials (25 ml) and were separated by taxonomic level by vial. We attempted to obtain at least 4 replicates (*i.e.,* 4 samples from each of the 2 spring sites) of all insect species when possible; however,

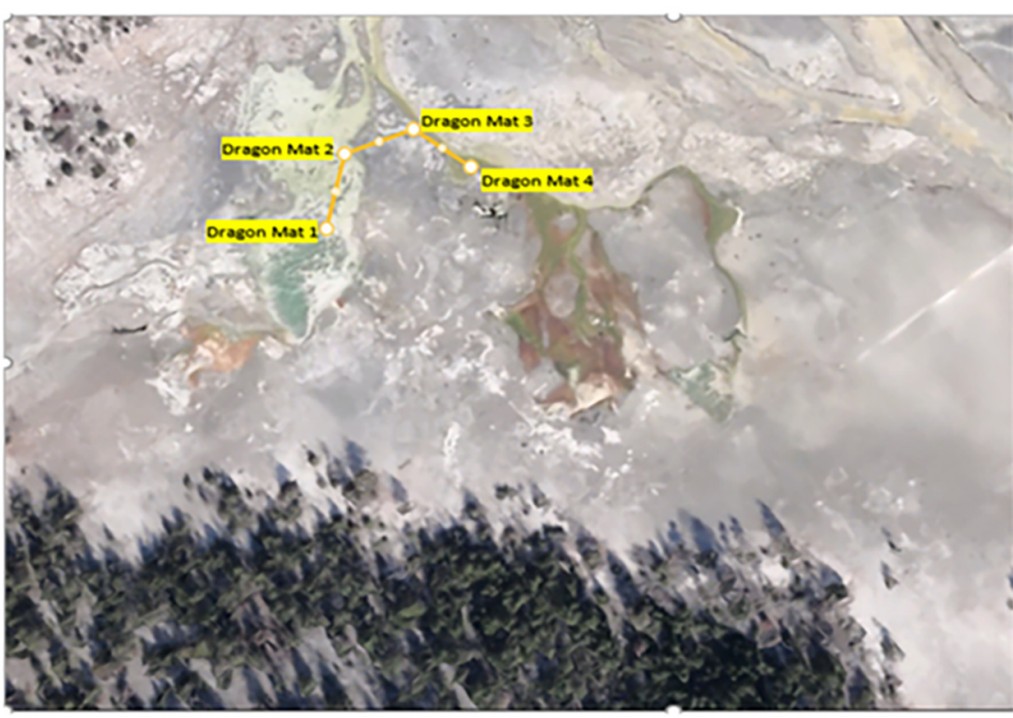

**Figure 2  Aerial view of Dragon-Beowulf Hot Springs and sampling locations.** The sampling pattern and locations are indicated along the path.

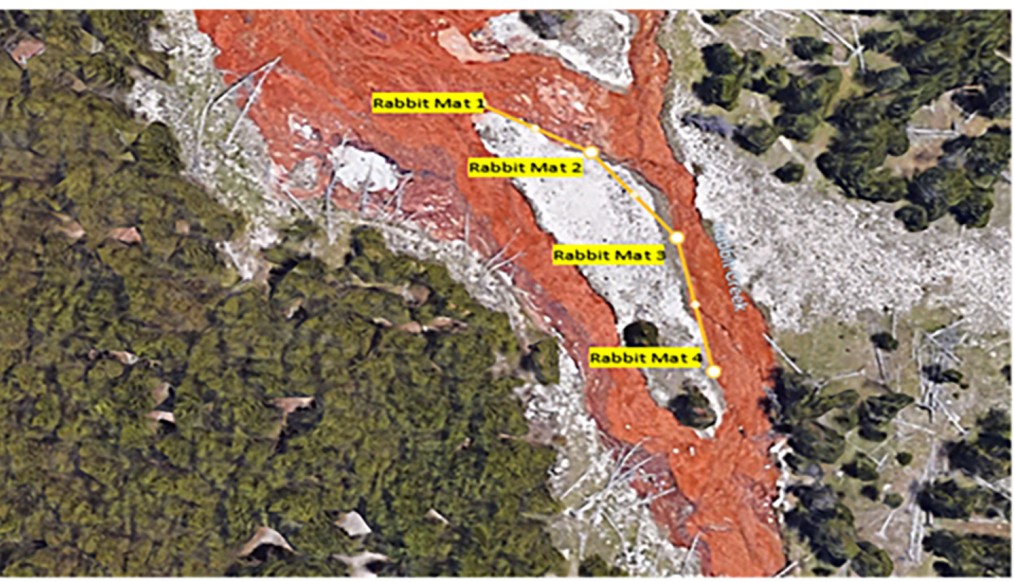

**Figure 3  Aerial view of rabbit creek hot springs and sampling locations.** Sampling locations are designated along the path.

certain taxa were absent on some sampling dates and some locations. Except for tiger beetles and adult stratiomyids, insects were pooled to meet mass requirements (ca. 0.005 g) for inductively coupled plasma mass spectrometry metal analysis, consequently $N = 1$ for each insect sample location-year combination.

## Sample preparation and analysis

Tissue processing for elemental analysis was modified from procedures used by used by *Alowaifeer et al. (2023)* and by *Gotschall (2021)*. Individuals were pooled into single samples to provide sufficient mass for analysis. Specifically, in 2021 individuals/sample included 8 *C. haemorrhagica* each for Dragon-Beowulf Hot Springs and Rabbit Creek Hot Springs, 10 *A. mormon* at Rabbit Creek Hot Springs, 43 *P. turbida* at Dragon–Beowulf Hot Springs, 31 *P. turbida* at Rabbit Creek Hot Springs, 24 *R. distincta* at Rabbit Creek Hot Springs, seven *Nebria* sp. at Dragon-Beowulf Hot Springs, five *Odontomyia* larvae at Dragon-Beowulf Hot Springs, 2 *Odontomyia* adults at Dragon-Beowulf Hot Springs, and 76 *S. littoralis* at Dragon-Beowulf Hot Springs. In 2022, individuals/sample included 8 *C. haemorrhagica* each for Dragon-Beowulf Hot Springs and Rabbit Creek Hot Springs, 15 *A. mormon* at Rabbit Creek Hot Springs, 38 *P. turbida* at Dragon-Beowulf Hot Springs, 22 *P. turbida* at Rabbit Creek Hot Springs, 11 *R. distincta* at Rabbit Creek Hot Springs, 20 *Nebria* sp. at Dragon-Beowulf Hot Springs, and 20 *S. littoralis* at Dragon-Beowulf Hot Springs.

Insect biomass was digested by mixing with 2.0 ml of concentrated HCl and 2.0 ml of concentrated $HNO_3$ and then heating (98 °C for 25 min at ambient air pressure) to dissolve sample tissues. After cooling, 16.0 ml of distilled water was then added to each vial (20 ml total for each sample). Eight ml of each sample was placed in a new vial for total metal analysis using an Agilent 7800 Inductively Coupled Plasma Mass Spectrometer, reaction cell ($H_2$ and He mode capability) with upper and lower limits of quantification of ca. 0.5 and 500 ng metal/g tissue. All metals analyzed were above the lower limits of quantification. Lower limits of quantification were calculated from the mean of metal-specific blank samples plus three standard deviations multiplied by 8 (analytical samples were based on 8 ml aliquots of acid digested tissue). Certified standards from CPI International (Product # 440 –121116NC02) were used for generation of calibration curves and check standards for the quantifications.

Additional analysis for total and methylated mercury was conducted by Brooks Applied Labs (BAL; Seattle, Washington) for a limited number of pooled samples (see https://brooksapplied.com for more information on methods). Quoting their procedures for methylated mercury: "All tissue samples for MeHg analysis were extracted with a mixture of potassium hydroxide and methanol in accordance with BAL SOPs [standard operating procedures]. Extracts were then analyzed *via* cold vapor gas chromatography atomic fluorescence spectroscopy (CV-GC-AFS)". For total mercury "All tissue samples for Hg analysis were digested *via* modified EPA Method 1631, Appendix using a mixture of concentrated nitric acid and concentrated sulfuric acid. The digested samples were preserved with bromine monochloride prior to analysis. The preserved digests were then analyzed *via* cold vapor atomic fluorescence spectroscopy (CVAFS)". Depending on the specific sample, the minimum detection limit ranged between 0.351 to 7.4 ng/g.
All tissue concentrations are reported on a wet-weight basis. For conversion to dry weight a standard conversion factor for fish and invertebrate tissue is 0.2 (*Meador, 2011*). This conversion was used by *Rodriguez et al. (2018)* for invertebrates, and we adjusted HC50 values from *Rodriguez et al. (2018)* to wet-weight for comparison to our wet-weight values. Rodriguez et al.'s data are based on Spanish streams of varying pH, size, and associations with traditional mining operations. Given the diversity of stream conditions in *Rodriguez et al. (2018)* and focus on tissue concentrations of metals, we believe their HC50 values provide a suitable comparison for streams in this study.

Descriptive statistics (means and standard errors) were calculated in Excel 365 where possible.

## RESULTS AND DISCUSSION

Tables 1 and 2 list the species collected at both sites in 2021 and 2022. *Paracoenia turbida* occurred at both Dragon-Beowulf Hot Springs and Rabbit Creek Hot Springs and based on our observations represent the second trophic level, *i.e.,* consumers of the algal/bacterial mats. Similarly, at Dragon-Beowulf Hot Springs, *Odontomyia* sp. also represent the second trophic level. Predaceous (tertiary trophic level) insect species found at Dragon-Beowulf Hot Springs were *Nebria* sp. and *S. littoralis*. Predaceous (tertiary trophic level) insects found at Rabbit Creek Hot Springs were *A. mormon*, and *R. disticta champion*. The apex invertebrate predator at both sites seems to be the tiger beetle *C. haemorrhagica*. *Cicindelidia haemorrhagica* is the largest insect predator in these two hot springs communities and because it preys on *P. turbida* and other predator species (as well as various incidental species occurring in hot springs) it occupies a tertiary or quaternary trophic level. Additionally, our observations indicate *C. haemorrhagica* foraged over the greatest area (aquatic and terrestrial) of any other predator (*Gotschall, 2021*; *Willemssens, 2019*).

During sampling, adult and nymphal *S. littoralis* were found exclusively in association with larval *P. turbida* at Dragon-Beowulf Hot Springs. At Rabbit Creek Hot Springs, although the primary potential prey for naucorids and veliids also was *P. turbida*, unlike *S. littoralis* the naucorids and veliids foraged beyond areas limited to *P. turbida*. Stratiomyid larvae occurred only at Dragon-Beowulf Hot Springs and were sampled from within an algal mat that contained *P. turbida*.

Concentrations of 12 heavy metals in sampled taxa in both locations are presented in Tables 1 and 2 for 2021 and 2022, respectively. Additionally, Table 3 presents concentrations of total and methyl mercury for selected species in 2022 and in 2021 for *Odontomyia* sp. Concentrations of mercury show general agreement (within an order of magnitude) for tested species, despite differences in analytic techniques (ICAP mass spectroscopy and cold vapor gas chromatography atomic fluorescence spectroscopy) (Tables 2 and 3).

Metal concentrations within species varied by location that derive from fundamentally different geologic origins yielding significant different levels of pH and solutes, including the metal(loid)s targeted in this study (Phelp and Buseck 1980). At a general level, Norris Geyser Basin is exceedingly heavily enriched with mercury (Phelp and Buseck 1980), providing a background level orders of magnitude greater than Rabbit Creek. This difference is reflected

**Table 1 Concentration of six heavy metals in insect taxa.** Means and standard errors (SE) of wet weight concentrations (ng metal/g tissue) of six heavy metals (chromium, manganese, cobalt, nickel, copper, zinc, and arsenic) among insect taxa sampled in 2021 and 2022 from two thermal features, Dragon-Beowulf Hot Springs (an acidic spring) and Rabbit Creek Hot Springs (an alkaline spring) in Yellowstone National Park, as determined by inductively coupled plasma mass spectrometry in 2021 and 2022. $N = 2$ for each species location (mean across years), except for *Odontomyia* sp. which were found only in 2021.

| Species | Statistic | Concentration (ng/g) | | | | | | |
| --- | --- | --- | --- | --- | --- | --- | --- | --- |
| | | Chromium | Manganese | Cobalt | Nickel | Copper | Zinc | Arsenic |
| **Dragon-Beowulf Hot Springs** | | | | | | | | |
| *Cicindelidia haemorrhagica* | mean | 944 | 84.8 | 15.9 | 578 | 40.2 | 210 | 3.94 |
| | SE | 18.7 | 1.34 | 0.21 | 8.13 | 0.21 | 6.36 | 0.00 |
| *Salda littoralis* | mean | 754 | 325 | 5.32 | 380 | 222 | 3,270 | 633 |
| | SE | 478 | 206 | 3.38 | 241 | 141 | 2,070 | 401 |
| *Nebria* sp. | mean | 377 | 135 | 2.90 | 194 | 39.0 | 1,910 | 13.0 |
| | SE | 215 | 77.0 | 1.65 | 111 | 22.3 | 1,090 | 7.39 |
| *Paracoenia turbida* | mean | 1,001 | 59.4 | 5.48 | 460 | 88.4 | 318 | 508 |
| | SE | 141 | 8.41 | 0.78 | 65.1 | 12.5 | 44.9 | 71.8 |
| *Zygogonium* sp. | mean | 2.89 | 1.39 | 0.07 | 1.93 | 1.41 | 3.66 | 29.0 |
| | SE | 0.88 | 0.62 | 0.03 | 0.57 | 0.51 | 1.47 | 8.10 |
| *Odontomyia* sp.–larvae | $n = 1$ | 0.21 | 0.08 | BDL | 0.11 | 0.02 | 1.09 | 0.01 |
| *Odontomyia* sp.–adult | $n = 1$ | 0.14 | 0.01 | BDL | 0.07 | 0.01 | 0.04 | 0.07 |
| **Rabbit Creek Hot Springs** | | | | | | | | |
| *Cicindelidia haemorrhagica* | mean | 65.1 | 34.3 | 1.06 | 31.8 | 57.6 | 1,270 | 9.09 |
| | SE | 1.73 | 3.78 | 0.05 | 0.46 | 2.12 | 2,550 | 0.72 |
| *Rhagovellia distincta* | mean | 3,038 | 387 | 24.5 | 1,580 | 420 | 2,440 | 55.5 |
| | SE | 1,490 | 190 | 12.0 | 777 | 207 | 1,200 | 27.3 |
| *Ambrysus mormon* | mean | 34.4 | 26.0 | 0.44 | 16.2 | 21.3 | 214 | 25.3 |
| | SE | 18.1 | 13.7 | 0.23 | 8.53 | 11.2 | 112 | 13.3 |
| *Paracoenia turbida* | mean | BDL | BDL | BDL | BDL | BDL | BDL | 0.03 |
| | SE | BDL | BDL | BDL | BDL | BDL | BDL | 0.16 |
| cyanobacteria | mean | 0.47 | 0.04 | 0.01 | 0.29 | 0.02 | 0.11 | 0.00 |
| | SE | 1.96 | 2.35 | 0.23 | 1.00 | 1.63 | 7.24 | 0.64 |

in ~30-fold difference in mercury levels in the Cyanidioschyzon algae at Dragon-Beowulf *versus* Cyanobacteria at Rabbit Creek (Table 2). Of note regarding potential prey availability, given its location within the Norris Geyser Basin, Dragon-Beowulf Hot Springs is a largely depauperate environment with few adjacent plants and limited animal species.

In contrast, Rabbit Creek Hot Springs is a thermal creek in a high valley, surrounded by grasses and forest. Consequently, Rabbit Creek Hot Springs potentially features greater number and diversity of transient taxa. If predator species prey on these non-resident organisms, the predators should have reduced heavy metal concentration. Variability in metal concentrations measured between the primary consumer (*P. turbida*) and apex invertebrate predator (*C. haemorrhagica*) is consistent with dietary differences, specifically feeding by *C. haemorrhagica* on insects from outside the thermal areas.

In characterizing food chains, the algal and bacterial mats provide the base level of the food chain and that brine flies feed on these mats. In the absence of other potential prey,

**Table 2** Means and standard errors (SE) of wet weight concentrations (ng metal/g tissue) of six heavy metals (selenium, cadmium, silver, tin, antimony, mercury, and lead) among insect taxa sampled in 2021 and 2022 from two thermal features, Dragon-Beowulf Hot Springs (an acidic spring) and Rabbit Creek Hot Springs (an alkaline spring) in Yellowstone National Park, as determined by inductively coupled plasma mass spectrometry in 2021 and 2022. *N = 2* for each species location (mean across years), except for *Odontomyia* sp. which were found only in 2021.

| Species | Stat | Selenium | Cadmium | Silver | Tin | Antimony | Mercury | Lead |
|---|---|---|---|---|---|---|---|---|
| | | | | **Concentration (ng/g)** | | | | |
| | | | | **Dragon-Beowulf Hot Springs** | | | | |
| *Cicindelidia haemorrhagica* | mean | 0.44 | 0.30 | 1.06 | 10,850 | 0.63 | 0.06 | 1.50 |
| | SE | 0.00 | 0.07 | 0.00 | 247 | 0.01 | 0.00 | 0.01 |
| *Salda littoralis* | mean | 2.05 | 1.11 | 12.02 | 85,400 | 0.94 | 1.47 | 5.80 |
| | SE | 1.30 | 0.71 | 7.62 | 54,200 | 0.60 | 0.93 | 3.68 |
| *Nebria sp.* | mean | 0.15 | 0.22 | 0.42 | 34,900 | 0.47 | 0.06 | 1.87 |
| | SE | 0.08 | 0.12 | 0.24 | 19,900 | 0.27 | 0.03 | 1.07 |
| *Paracoenia turbida* | mean | 0.66 | 0.51 | 0.93 | 80,850 | 1.23 | 0.53 | 2.20 |
| | SE | 0.09 | 0.09 | 0.13 | 11,400 | 0.17 | 0.07 | 0.31 |
| *Zygogonium* sp. | mean | 0.01 | 0.01 | 0.01 | 708 | 1.19 | 0.87 | 0.20 |
| | SE | 0.00 | 0.00 | 0.00 | 220 | 0.40 | 0.24 | 0.06 |
| *Odontomyia* sp.–larvae | *n = 1* | 0.09 | 0.02 | 0.08 | 6,270 | 1.28 | 0.57 | 1.20 |
| *Odontomyia* sp.–adult | *n = 1* | 0.05 | 0.06 | 0.01 | 973 | 0.17 | 0.02 | 0.67 |
| | | | | **Rabbit Creek Hot Springs** | | | | |
| *Cicindelidia haemorrhagica* | mean | 0.78 | 0.45 | 1.73 | 13,400.00 | 0.17 | 0.20 | 5.33 |
| | SE | 0.04 | 0.06 | 0.04 | 1,100 | 0.01 | 0.01 | 0.84 |
| *Rhagovellia distincta* | mean | 7.50 | 10.9 | 15.8 | 602,000 | 6.67 | 0.52 | 5.25 |
| | SE | 3.68 | 5.47 | 7.74 | 296,000 | 3.27 | 0.25 | 2.58 |
| *Ambrysus mormon* | mean | 1.09 | 0.67 | 0.80 | 4,320 | 1.15 | 0.05 | 0.47 |
| | SE | 0.57 | 0.35 | 0.42 | 2,280 | 0.61 | 0.03 | 0.25 |
| *Paracoenia turbida* | mean | 0.97 | 0.95 | 0.67 | 63,000 | 1.64 | 0.10 | 1.57 |
| | SE | 0.03 | 0.03 | 0.02 | 16,600 | 0.04 | 0.00 | 0.04 |
| cyanobacteria | mean | 0.03 | 0.01 | 0.01 | 528 | 1.06 | 0.03 | 0.15 |
| | SE | 0.01 | 0.00 | 0.01 | 177 | 0.32 | 0.01 | 0.05 |

predatory species, other than *C. haemorrhagica,* are limited to *P. turbida*. The diet of *C. haemorrhagica* is interesting because their foraging in thermal pools (some as hot as 70 °C!) seems driven by scavenging on flying species caught by the heat. Thus, *C. haemorrhagica*'s is likely a combination of *P. turbida*, small predators, and some proportion of food from outside the thermal areas.

With respect to mercury, which is of ecological concern, cyanobacterial mats showed only about 0.5% methylated mercury. However, all total mercury in *C. haemorrhagica* samples was methylated. These data are consistent with the notion that at Dragon-Beowulf Hot Springs and Rabbit Creek Hot Springs the primary biological mobility of mercury is in the methylated form.

One major observation from this study was the difference in metal concentrations reflected between the larval and adult Stratiomyidae. Declines across multiple metals were seen with the most notable reductions in antimony (87% less in adults than larvae), mercury (97% less in adults than larvae), and arsenic (99% less in adults than larvae) (Tables 1 and

**Table 3  Concentrations of mercury in selected taxa.** Wet weight concentrations (as determined by cold vapor gas chromatography atomic fluorescence spectroscopy) of total mercury (ng total Hg/g tissue) and methyl mercury MeHg in selected species sampled in 2022, except for the *Odontomyia* samples, which were obtained in 2021. $N = 1$ for each species sample year-location combination (pooled samples).

| Source/species | ng/g | form of Hg |
|---|:---:|---|
| **Dragon-Beowulf Hot Springs** | | |
| *Cyanidioschyzon* sp. mat | 0.0251 | MeHg |
| *Odontomyia* sp. | 0.676 | Total Hg |
| *Odontomyia* sp. | 0.047 | MeHg |
| *Cicindelidia haemorrhagica* | 0.113 | MeHg |
| *Cicindelidia haemorrhagica* | 0.305 | Total Hg |
| **Rabbit Creek Hot Springs** | | |
| Cyanobacterial mat | 0.633 | Total Hg |
| Cyanobacterial mat | 0.0034 | MeHg |
| *Cicindelidia haemorrhagica* | 0.0954 | Total Hg |
| *Cicindelidia haemorrhagica* | 0.0994 | MeHg |
| *Ambrysus mormon* | 0.0195 | MeHg |

2). Flies (midges) in human-impacted streams retained heavy metals including mercury due to presumed efficient transfer between molts (*Marziali, Roscioli & Valsecchi, 2021*).

Because only adult ephydrids were collected during this study, it is possible that these flies accumulate high concentrations of heavy metals as larvae. If so, they may be sequestering these metals into their cuticle, and as they pupate metals remain in the exuviae. This mechanism of detoxification has been documented in other insects (*Marziali, Roscioli & Valsecchi, 2021*; *Kraus et al., 2021*). In particular, *Kraus et al. (2021)* noted that cadmium, copper, zinc, and lead concentrations in stoneflies, mayflies, aquatic Diptera, and caddisflies were reduced in adult forms as compared to nymphal and larval insects.

Alternatively, the large difference between total and methylated mercury concentrations might imply that mercury deposited in the cuticle is de-methylated (Table 3). Specifically, in larval *Odontomyia* sp. 0.676 ng/g total mercury was observed *versus* 0.047 ng/g methyl mercury (Table 3). In contrast, all mercury in *C. haemorrhagica* was methylated. Thus, if the primary mobility of mercury through tropic levels is in the methylated form, high levels of non-methylated mercury in *Odontomyia* sp. seem likely to be associated with either detoxification and/or sequestration. As shown with *Odontomyia* sp., sequestering metals into the cuticle might be common among holometabolous insects at these streams. For example, *Gotschall (2021)* found that *C. haemorrhagica* sequestered heavy metals in its exoskeleton, with metals particularly accumulated in abdominal sternites. For these reasons, further studies examining both larval and pupal stages of holometabolous insects and nymphs of hemimetabolous groups could add to our understanding of heavy metal detoxification and tolerance among insects occurring in these thermal springs.

Bioaccumulation factors for sampled insects show accumulation within taxa and in some instances biomagnification across the trophic webs; patterns were consistent across years (Tables 4 and 5). Evidence of biomagnification was seen for multiple metals and in multiple insect species. At least two orders of magnitude increases were seen in *P. turbida*

**Table 4  Bioaccumulation factors for metals from Table 1.** Means and standard errors (SE) of bioaccumulation factors (BAF) based on wet weight concentrations in Table 1, with the BAF equal to concentration within species divided by concentration within mat for Dragon-Beowulf Hot Springs and Rabbit Creek Hot Springs in 2021 and 2022. Data rounded to three significant digits. $N = 2$ for each species location (mean across years), except for *Odontomyia* which were found only in 2021.

| | | Bioaccumulation factors based on mat concentration | | | | | | |
|---|---|---|---|---|---|---|---|---|
| **Species** | **Stat** | **Chromium** | **Manganese** | **Cobalt** | **Nickel** | **Copper** | **Zinc** | **Arsenic** |
| | | **Dragon-Beowulf Hot Springs** | | | | | | |
| *Cicindelidia haemorrhagica* | mean | 405 | 103 | 311 | 367 | 39.1 | 87.0 | 0.2 |
| | SE | 129 | 47.2 | 123 | 112 | 14.4 | 36.8 | 0.0 |
| *Salda littoralis* | mean | 197 | 169 | 52.3 | 149 | 116 | 646 | 16.7 |
| | SE | 106 | 73.0 | 25.4 | 81.0 | 58.5 | 306 | 9.2 |
| *Nebria* sp. | mean | 215 | 245 | 81.0 | 163 | 53.3 | 1120 | 0.7 |
| | SE | 140 | 165 | 53.8 | 106 | 35.2 | 748 | 0.4 |
| *Paracoenia turbida* | mean | 462 | 80.5 | 118 | 314 | 94.2 | 143 | 22.5 |
| | SE | 188.8 | 42.1 | 56.1 | 126 | 43.0 | 69.6 | 8.7 |
| *Zygogonium* sp. | mean | 1.0 | 1.0 | 1.0 | 1.0 | 1.0 | 1.0 | 1.0 |
| | SE | 0.0 | 0.0 | 0.0 | 0.0 | 0.0 | 0.0 | 0.0 |
| *Odontomyia* sp.–larvae | $n = 1$ | 13.0 | 20.1 | 11.6 | 21.3 | 41.7 | 17.6 | 13.6 |
| *Odontomyia* sp.–adult | $n = 1$ | 4.4 | 29.1 | 3.7 | 3.9 | 15.8 | 163 | 0.2 |
| | | **Rabbit Creek Hot Springs** | | | | | | |
| *Cicindelidia haemorrhagica* | mean | 18.3 | 7.4 | 2.6 | 17.4 | 18.1 | 305 | 6.3 |
| | SE | 7.3 | 3.1 | 1.1 | 6.9 | 7.0 | 0.0 | 2.4 |
| *Rhagovellia distincta* | mean | 522 | 50.4 | 35.8 | 534 | 82.1 | 109 | 24.7 |
| | SE | 91.9 | 10.7 | 6.2 | 92.6 | 16.2 | 20.7 | 5.9 |
| *Ambrysus mormon* | mean | 13.3 | 7.2 | 1.4 | 12.4 | 9.0 | 20.9 | 22.4 |
| | SE | 8.6 | 4.6 | 0.9 | 8.0 | 5.8 | 13.5 | 14.2 |
| *Paracoenia turbida* | mean | 175 | 9.6 | 8.7 | 168 | 17.4 | 18.0 | 3.9 |
| | SE | 63.6 | 3.2 | 3.1 | 61.3 | 6.0 | 6.3 | 1.2 |
| cyanobacteria | mean | 1.0 | 1.0 | 1.0 | 1.0 | 1.0 | 1.0 | 1.0 |
| | SE | 0.0 | 0.0 | 0.0 | 0.0 | 0.0 | 0.0 | 0.0 |

for all metals except antimony, mercury, and lead, with the highest bioaccumulation factor (729) for chromium. At the other end of the food web, the apex invertebrate predator, *C. haemorrhagica*, had at least 10-fold BAF for all metals at some location-year combinations except with antimony. In *C. haemorrhagica* the greatest BAFs occurred with chromium (588), cobalt (485), nickel (526), and zinc (1,740). Of other taxa, high BAFs were observed for zinc with *Nebria* sp. (2,180) and for *S. littoralis* (1,080).

Regarding specific metals, because of their toxicity and potential for movement in terrestrial food webs, arsenic and mercury are of particular interest. Several forms of arsenic occur in natural waters, depending upon the redox potential and pH (*Inskeep, McDermott & Fendorf, 2002*; *McCleskey et al., 2010*), the two most common species being arsenic (III) and arsenic (V). Both arsenic (III) and arsenic (V) form stable bonds with carbon, resulting in numerous organo-arsenic compounds, some of which can be very toxic (*Genchi et al., 2022*).

**Table 5 Bioaccumulation factors for metals from Table 2.** Means and standard errors (SE) of bioaccumulation factors (BAF) based on wet weight concentrations in Table 2, with the BAF equal to concentration within species divided by concentration within mat for Dragon-Beowulf Hot Springs or Rabbit Creek Hot Springs in 2021 and 2022. Data rounded to three significant digits; $N = 2$ for each species sample location.

| | | Bioaccumulation factors based on mat concentration | | | | | | |
|---|---|---|---|---|---|---|---|---|
| Species | Stat | Selenium | Cadmium | Silver | Tin | Antimony | Mercury | Lead |
| | | **Dragon-Beowulf Hot Springs** | | | | | | |
| *Cicindelidia haemorrhagica* | mean | 92.3 | 66.8 | 186 | 19.2 | 0.7 | 0.1 | 9.1 |
| | SE | 30.9 | 24.2 | 60.1 | 6.3 | 0.2 | 0.0 | 2.7 |
| *Salda littoralis* | mean | 247 | 187 | 1240 | 90.6 | 0.6 | 1.3 | 21.9 |
| | SE | 129 | 113 | 655 | 48.4 | 0.3 | 0.7 | 11.8 |
| *Nebria sp.* | mean | 42.5 | 53.7 | 101 | 82.8 | 0.7 | 0.1 | 15.4 |
| | SE | 27.8 | 32.1 | 66.2 | 53.9 | 0.5 | 0.1 | 10.0 |
| *Paracoenia turbida* | mean | 150 | 112 | 178 | 154 | 1.5 | 0.8 | 14.6 |
| | SE | 64.8 | 34.9 | 75.9 | 64.2 | 0.6 | 0.3 | 6.0 |
| *Zygogonium* sp. | mean | 1.0 | 1.0 | 1.0 | 1.0 | 1.0 | 1.0 | 1.0 |
| | SE | 0.0 | 0.0 | 0.0 | 0.0 | 0.0 | 0.0 | 0.0 |
| *Odontomyia* sp.–larvae | $n = 1$ | 26.1 | 5.87 | 19.6 | 15.8 | 2.06 | 1.07 | 10.5 |
| *Odontomyia* sp.–adult | $n = 1$ | 14.1 | 16.3 | 2.94 | 2.45 | 0.27 | 0.03 | 5.87 |
| | | **Rabbit Creek Hot Springs** | | | | | | |
| *Cicindelidia haemorrhagica* | mean | 42.7 | 52.8 | 167 | 34.5 | 0.2 | 9.7 | 52.4 |
| | SE | 18.0 | 22.5 | 63.9 | 13.6 | 0.1 | 4.1 | 24.1 |
| *Rhagovellia distincta* | mean | 245 | 778 | 952 | 985 | 5.4 | 14.6 | 30.2 |
| | SE | 41.7 | 206 | 183 | 230 | 1.4 | 2.6 | 6.3 |
| *Ambrysus mormon* | mean | 80.8 | 96.0 | 106 | 14.3 | 1.8 | 3.4 | 5.7 |
| | SE | 52.5 | 60.8 | 68.1 | 9.1 | 1.1 | 2.2 | 3.7 |
| *Paracoenia turbida* | mean | 50.3 | 98.8 | 62.3 | 151 | 1.9 | 4.3 | 13.6 |
| | SE | 18.5 | 29.8 | 21.7 | 47.4 | 0.5 | 1.6 | 4.5 |
| cyanobacteria | mean | 1.0 | 1.0 | 1.0 | 1.0 | 1.0 | 1.0 | 1.0 |
| | SE | 0.0 | 0.0 | 0.0 | 0.0 | 0.0 | 0.0 | 0.0 |

The arsenic in YNP thermal waters is largely derived from deep thermal fluids, where water-rock interactions control surficial compositions (*McCleskey et al., 2010*; *USGS, 2021*). Arsenic concentrations in the main rivers draining YNP are often reported as elevated due to little arsenic being lost over long distances (*USGS, 2021*). The arsenic species found in measurements taken from the Gibbon River and inflows near its confluence (below Dragon-Beowulf Hot Springs) existed primarily as arsenic (V) (*McCleskey et al., 2010*).

Biomagnification of arsenic in the aquatic food chain is not frequent (*Rahman, Hasegawa & Lim, 2012*). Biomagnification has been reported in fish and gastropods, but mainly as AsB (arsenobetaine), which is rapidly metabolized, eliminated, and less toxic, therefore, posing less of an environmental hazard (*Maher et al., 1999*; *Caussy et al., 2003*). Although biomagnification of arsenic is uncommon, we observed five instances of BAFs of arsenic near or exceeding 10: *Odontomyia* sp. larvae (13.6), *P. turbida* (34.8), *C. haemorrhagica* (9.7), *R. distincta* (16.3), and *A. mormon* (42.8). If arsenic does routinely accumulate in these species, it presents a potential risk for other species higher in the food chain.

It is estimated that more than 90% of the mercury derived from anthropogenic sources and released to the atmosphere in the last 100 years is absorbed in the terrestrial environment with one-third of the total mercury in global fluxes estimated to arise from natural emissions (*Richardson et al., 2003*). Mercury is strongly retained in watersheds (*Mason, Laporte & Andres, 2000*) and can exist in many forms—the two which pertain most to our study sites are inorganic (Hg(II)) and methylmercury.

Methylmercury has the potential for mercury contamination to become an even more serious environmental health issue. Research suggests that as inorganic mercury settles into the sediments of aquatic systems, dominant anaerobic microorganisms are largely responsible for this methylation process. When the location in which mercury deposits is favorable, its conversion from inorganic mercury to the more toxic organic form occurs (*Tshumah-Mutingwende & Takahashi, 2019*). Both forms can be absorbed but inorganic mercury is generally taken up at a slower rate and with lower efficiency than the methylated form (*Ma, Du & Wang, 2019*). Mercury sources at YNP are unique in that not only are there potential contributions from regional coal-burning power plants and wildfires, but also from the many natural geothermal features throughout YNP. Studies done within YNP found that mercury commonly occurred in its methylated form (*King et al., 2006*) and that insects are potentially important vectors of Hg transfer into terrestrial food chains (*Boyd et al., 2009*). Our results (Table 3) are consistent with *Boyd et al. (2009)* findings regarding mercury at Dragon-Beowulf Hot Springs and Rabbit Creek Hot Springs.

Are concentrations of heavy metals in thermal environments of YNP toxic (acutely or chronically) to insects? Answering this question is complicated by the longstanding problem in relating aquatic concentrations of metals to tissue-based concentrations (*e.g.*, *Adams et al., 2010*; *Flanders et al., 2019*). Despite this dilemma, criteria for potential acute toxicity of metals to insects have emerged, including the HC50—median hazard concentration (*Rodriguez et al., 2018*) and methyl mercury concentrations (*Flanders et al., 2019*). These values are derived from watersheds homologous in structure and function to those in this study and are based on tissue concentrations, suggesting their potential usefulness in various systems. Table 6 compares metal concentrations in *C. haemorrhagica* to these two criteria and indicates that no metals occur at potentially toxic levels. These indications of toxicity are based on acute toxicity and single metals; chronic toxicity and potential synergistic effects remain uncertain.

If *C. haemorrhagica* experiences chronic toxicity from multiple metals, how does it avoid death? The most obvious answer and the one for which we have evidence is that *C. haemorrhagica* sequesters roughly 50% of the metals it accumulates in these streams into its cuticle (*Gotschall, 2021*). Based on our evidence of metal sequestration in *Odontomyia* sp., previous evidence with *C. haemorrhagica* (*Gotschall, 2021*), and studies with midges (*Marziali, Roscioli & Valsecchi, 2021*) and diverse aquatic insects (*Kraus et al., 2021*), sequestration of metals seems likely to be a common mechanism among the insect fauna of thermal pools in YNP. In contrast to work in other aquatic systems in which contaminants did not bioaccumulate in the adult stage (*Kraus et al., 2021*), insects of thermal pools in YNP had substantial bioaccumulation with specific metals. Unlike observations of aquatic insects in *Kraus et al. (2021)*, which had exposure to metals only in

 

**Table 6  Indications of toxicity to Cicindelidia haemorrhagica from elemental metals based on tissue-based toxicity criteria from two sources.** All concentrations are in ng/g with measured concentrations on a wet-weight basis. Criteria include the HC50 –median hazard concentration (*Rodriguez et al., 2018*) and methyl mercury [MeHg] concentration (*Flanders et al., 2019*). HC50 values converted to a wet weight basis assuming a 0.2 conversion factor (*Meador, 2011*). $N = 2$ for all means and standard errors (SE) (averaged across years).

| Criteria | As | Cd | Cu | Cr | Hg | Ni | Pb | Se | Zn |
|---|---|---|---|---|---|---|---|---|---|
| HC50 ng/g | 5,088 | 2,868 | 34,992 | 7,548 | 480 | 5,220 | 38,472 | 9,156 | 393,168 |
| [MeHg] | | | | | 102 | | | | |
| **Stat** | **As** | **Cd** | **Cu** | **Cr** | **Hg** | **Ni** | **Pb** | **Se** | **Zn** |
| | | | | **Dragon-Beowulf Hot Springs** | | | | | |
| Mean | 3.94 | 0.298 | 40.2 | 944 | 0.056 | 578 | 1.50 | 0.443 | 210 |
| SE | 0 | 0.07 | 0.212 | 18.7 | 0.001 | 8.13 | 0.011 | 0.004 | 6.36 |
| | | | | **Rabbit Creek Hot Springs** | | | | | |
| Mean | 9.08 | 0.454 | 57.6 | 65.05 | 0.204 | 31.8 | 5.33 | 0.78 | 5360 |
| SE | 0.718 | 0.064 | 2.12 | 1.73 | 0.014 | 0.46 | 0.841 | 0.041 | 2630 |

the aquatic immature stages, this occurrence is likely due to continual metal exposure as adults.

Overall, the evidence shows bioaccumulation of various heavy metals by insects in the thermal areas of Yellowstone, but the levels of bioaccumulation do not seem to represent acute risks to insect species. Chronic risks from heavy metals and potential synergies remain unclear, but evidence of insects sequestering metals might imply these risks are low. For mercury and arsenic the bioaccumulation data do suggest that insects might be a source of contamination for insectivorous animals feeding in the thermal areas.

## ACKNOWLEDGEMENTS

We appreciate the help and advice of T McDermott in conducting our metal assays, and the assistance of A Carlson, YNP Research Permit Office. We thank E Boyd, S Bradbury, and T McDermott for their extremely helpful reviews of drafts of this manuscript. We also thank B Schwartz. M Maxcer, M Lipinsky, and J Obafunwa for their work in the field.

### Funding

This study was supported with assistance from Therion, LLC, Brian Fiske, the Robert Allen Wright Endowment of Iowa State University, Montana State University, and the Montana Agricultural Experiment Station, and the University of Nebraska-Lincoln. There was no additional external funding received for this study. The funders had no role in study design, data collection and analysis, decision to publish, or preparation of the manuscript.

### Grant Disclosures

The following grant information was disclosed by the authors:

Therion, LLC, Brian Fiske, the Robert Allen Wright Endowment of Iowa State University, Montana State University, and the Montana Agricultural Experiment Station, and the University of Nebraska-Lincoln.

## Competing Interests

The authors declare there are no competing interests.

## Author Contributions

- Braymond Adams conceived and designed the experiments, performed the experiments, analyzed the data, prepared figures and/or tables, authored or reviewed drafts of the article, and approved the final draft.
- John Bowley performed the experiments, authored or reviewed drafts of the article, and approved the final draft.
- Monica Rohwer performed the experiments, authored or reviewed drafts of the article, and approved the final draft.
- Erik Oberg conceived and designed the experiments, authored or reviewed drafts of the article, and approved the final draft.
- Kelly Willemssens performed the experiments, authored or reviewed drafts of the article, and approved the final draft.
- Wendy Wintersteen conceived and designed the experiments, authored or reviewed drafts of the article, and approved the final draft.
- Robert K.D. Peterson conceived and designed the experiments, authored or reviewed drafts of the article, and approved the final draft.
- Leon G. Higley conceived and designed the experiments, analyzed the data, prepared figures and/or tables, authored or reviewed drafts of the article, and approved the final draft.

## Field Study Permissions

The following information was supplied relating to field study approvals (i.e., approving body and any reference numbers):

Studies were carried out under Yellowstone National Park Research Permits #8100 and #7092.

## Data Availability

The raw concentration values which are used in calculations of BAF values are available in the Supplementary File.

## Supplemental Information

Supplemental information for this article can be found online at http://dx.doi.org/10.7717/peerj.16827#supplemental-information.

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
