# Peer review of "Heavy metal movement through insect food chains in pristine thermal springs of Yellowstone National Park"

_PeerJ, doi:10.7717/peerj.16827_

## Round 0.1 · original submission · Major Revisions

Dear Dr. Higley,

after this first review round, one of the reviewers indicated minor reviews were needed, while the other indicated significant changes were needed.

In summary, improvements in your introduction, results, and discussion are needed. In the intro, you need to provide better support for using insects in this kind of study, along with minor improvements. In the results section, the main issue seems to be the lack of figures and the main use of tables to draw your main conclusions and interpretation of results. Please consider adding figures and other results that allow your readers to better understand what you want them to see. Finally, your discussion needs improvements to its end. More information and generalization of what exists in the literature is needed.

Please note that by the time you have a new version of your text, please do not forget to submit a rebuttal letter showing the main differences that you made and explaining those changes you did not agree with the reviewers.

Sincerely,
Daniel Silva

·

Basic reporting

The manuscript is well organized into the usual structure for a research manuscript, and the writing is generally clear and free from errors. I did not the following minor corrections or suggestions during my reading of the manuscript, and hope these will provide improvement of the final version.

Line 60: I think “a” should be inserted before “food web’s” in this sentence.
Line 139: Stratiomyidae is misspelled.
Line 143: Delete first comma, and “both” since more than 2 collection methods are listed.
Line 150: The first sentence here seems to belong on Line 136, to start the paragraph that reports how insects were identified. Also, why start this sentence with “macroinvertebrates” and not “insects”?
Line 172: I’d change “heated” to “heating” to match with “mixing” on the previous line and “cooling” on Line 173.
Line 182: “SOPs” needs to be spelled out somewhere so a reader can understand it.
Lines 196, 197, 199: “Rodriguez” is misspelled as “Rodrigez.”
Line 200: Change “provides” to “provide.”
Line 207: Change “represent” to “represents.”
Line 211: Is “champion” the author of this species? A subspecies? It’s italicized which signifies a Latin name.
Line 344: Delete the comma.


There are no figures, as all the data are summarized in a series of tables. The data tables were good, except I would recommend the authors consider significant digits in their reporting of mean and SE values. For example, in Table 1, the mean Zn concentration for Salda littoralis is 3268.50 with an SE value of 2072.88. These numbers suggest a precision of measurement of 5 or 6 significant digits for the original data and I am sure the technique used does not yield this level of precision on such low-mass samples. As a secondary issue, including all those digits makes numbers more difficult to read. I would suggest 2 (or maybe 3) significant digits for all the values reported in the manuscript to reflect measurement precision and also simplify understanding of the results. This issue is all the more important because the very low replication in this study, along with the large SE values for most data, tells us that the numbers reported are fuzzy estimates of “true” values.

I also have a few other points to make about specific tables:

In Table 2, headings are “Dragon Hot Springs” and “Rabbit Hot Spring” but these don’t match similar headings in all the other tables.

In Table 3, why was C. haemorrhagica MeHg concentration listed twice? Is one of those values supposed to be Total Hg?

In Table 5 (for which the header reads “Table 1”), the value for Ag for Salda littoralis runs into two lines since it has 6 characters and the column is apparently not wide enough for that.

Experimental design

The study design is reasonable given the objectives, and the contrast between the two spring systems (one very acidic and the other alkaline) is an interesting aspect of the research. Methods used to measure metals are appropriate, with inclusion of blanks and certified standards to help validate the measurements.

The main weakness of the study design is the low replication of samples. I recognize the difficulty of obtaining enough mass when dealing with small insects like the ones in this study, but two replicates is barely replication and can give only the most preliminary estimates of data points.

I also wonder: were any voucher samples of insects collected and deposited in a collection? This is a good practice so that future researchers can re-visit identifications since taxonomy is a “fluid” science and changes occur.

As a minor point, values of 0 for metal concentrations in Tables 1 and 2 probably should be listed instead as “B.D.L.” for “Below detection limits” (or some other acknowledgement that there probably were some atoms of those elements present in all samples but their level was not able to be documented with the methods used).

Validity of the findings

The research is directed toward a relatively novel system and this is a strength of the study. It also addresses an interesting ecological question with environmental significance, as transfer of heavy metals in food webs has implications for human health as well as the health of other organisms.

As I mention regarding the study design, low to no replication is a weakness. There is no statistical analysis of the data, just descriptive statistics using means and SE values. This means the research is descriptive but I think that given the special nature of these ecosystems it is worth reporting as an initial investigation that might generate more rigorous studies in the future.

Another potential problem with experimental technique that should be acknowledged by the authors is adhesion of dust or other external contamination to insect bodies. Was there any effort to examine samples for contaminants? Especially for any insects that are particularly hairy? Note that the literature on plant heavy metal concentrations contains reports of dust adhesion resulting in elevated values and, as a result, some recent research has involved washing samples to allow better documentation of plant tissue metal levels.

The discussion of the results ties the work into the literature in a reasonable way. By this I mean there are references that are relevant but not cited in the manuscript, yet I think their inclusion would bulk up the manuscript in a way that would not meaningfully contribute to the literature.

Additional comments

There is the matter of lack of information on exactly what each insect is eating. The authors assume the diet of all species in these ecosystems is based on the algal mats in the springs, but diet partly depends on the mobility of each species and its life stage. One example of this issue is the tiger beetle, which the authors state (lines 215-216) may prey on secondary or tertiary level species AND which forages over a wide area (adults are very mobile). This matters because BAF is calculated for each species based upon its (presumed) diet and the authors’ assumptions about diet are tenuous for at least some species. This means that there is considerable uncertainty about the accuracy of the major findings of the research. The authors address this factor on lines 240-246 of the discussion in the manuscript and I think they do a reasonable job of putting this matter forth for the reader to be aware of.

The authors should specify the life history stage of insects collected for all taxa. This was done for most of the insects in the methods, but not for taxa listed on lines 144-145.

Reviewer 2 ·

Basic reporting

The article is interesting and provides relevant data and is somewhat novel.

However, the introduction lacks in background information and also in stating the importance of the study. Nowhere is it mentioned that insects are actually the base of the food web of many other animals which means if the accumulate metals in this region it could be transferred to birds or lizards as well.

The results and discussion part ends very abruptly and there is no discussion on what the results actually mean in terms of should this be of concern or not and no indication of if the arsenic that accumulates could have any potential threats or effects on organisms in the environment other than the insects.

Some sentences are a bit confusing but can be easily fixed with some rephrasing. In general, the article lacks some information to make the it more cohesive and indicate the importance of the findings.

Experimental design

The experimental setup was good - a few suggestions have been added on the PDF in terms of adding tables ect. Furthermore, the number of times the sites were visited should be specified to add to the validity of the data - one sampling event in each month is not quite sufficient especially if it was only for an hour or two for example.

The research gap is stated well and answered however the implications of the results in a broader scope or ecological sense is lacking.

Validity of the findings

It was odd that no graphs where included not even to see of there was a trend in concentrations recorded in the collected tiger beetle between 2021 and 2022 for example , however the statistics that were done are sufficient.

The results and discussion part ends very abruptly and there is no discussion on what the results actually mean in terms of should this be of concern or not and no indication of if the arsenic that accumulates could have any potential threats or effects on organisms in the environment other than the insects. No mention to the original aim of the paper was made in the final conclusions.

Annotated reviews are not available for download in order to protect the identity of reviewers who chose to remain anonymous.

---

## Round 0.2 · Minor Revisions

Dear Dr. Higley,

In this new review round, one of your previous reviewers indicated minor reviewers were required. Still, some improvement concerning the manuscript figures and other minor issues are required. Please prepare a new version of the manuscript, correcting these issues. Specifically, take special care to improve the figures as requested by the reviewer.

Sincerely,
Daniel Silva

Reviewer 2 ·

Basic reporting

In general, the suggested revisions were made and has increased the value of the paper. However, suggestions to add graphs or some visual representation of the data has not been included, which is a shame as this would have made the paper much better. The suggested information that should have been added to the paper has been done in a minimalist way with no examples of metal movement from insects into the higher food chain which would have made the aim of this paper and importance of it more clear. Also no map of the study site was added, which would give a better idea of the layout of the experimental design than describing everything in a paragraph.

Some concerns are:

Line 53 which was added has no bearing on the paragraph and is not integrated well and there is also no reference for this sentence.

Line 86 should be rephrased to integrate this better into the previous paragraph and a reference should be added.

Line 89 - remove the word communities as the insects collected are hardly representative of the diversity that exists and community refers to perhaps doing a PCA or NMDS based on the metals accumulated in the insects at each sit to see if a particular metal may have influences some species at one of the sites more than the other site

Line 130 - numbers under 10 should be written out technically

Line 320-321 - Are there any examples that arsenic does travel p the food chain or at what concentration is could be potentially risky for insects i.e. what is the baseline number for arsenic?

Experimental design

No comment - see above

Validity of the findings

No comment - see above

---

## Round 0.3 · accepted · Accept

Dear Dr. Higley,

I am pleased to inform you that your manuscript has been formally accepted for publication in PeerJ. Congratulations!

Sincerely,
Daniel Silva